# Three Novel Cheiroid Hyphomycetes in *Dictyocheirospora* and *Dictyosporium* (*Dictyosporiaceae*) from Freshwater Habitats in Guangdong and Guizhou Provinces, China

**DOI:** 10.3390/jof10040259

**Published:** 2024-03-28

**Authors:** Yong-Xin Shu, Mingkwan Doilom, Saranyaphat Boonmee, Biao Xu, Wei Dong

**Affiliations:** 1Innovative Institute for Plant Health/Key Laboratory of Green Prevention and Control on Fruits and Vegetables in South China, Ministry of Agriculture and Rural Affairs, Zhongkai University of Agriculture and Engineering, Guangzhou 510225, China; shuyx799@163.com (Y.-X.S.); j_hammochi@hotmail.com (M.D.); biaoxu2008@hotmail.com (B.X.); 2Center of Excellence in Fungal Research, Mae Fah Luang University, Chiang Rai 57100, Thailand; saranyaphat.boo@mfu.ac.th; 3School of Science, Mae Fah Luang University, Chiang Rai 57100, Thailand

**Keywords:** aquatic environment, cheirosporous, phylogeny, taxonomy, three new species

## Abstract

Over the past two decades, numerous novel species have been identified within *Dictyosporiaceae*, primarily in *Dictyocheirospora* and *Dictyosporium*. A recent monograph has revealed that these two genera exhibit a distinct preference for freshwater habitats, particularly in southern China. However, further investigation into the distribution and diversity of the two genera in Guangdong and Guizhou Provinces remains insufficient. In this study, we conducted an analysis of four intriguing cheiroid hyphomycetes collected from flowing rivers in these two regions. Through morphological and phylogenetic analyses incorporating combined LSU, SSU, ITS, and *tef1-α* sequence data, we have identified them as a novel species in *Dictyocheirospora* (*Dictyoc. submersa* sp. nov.), two novel species in *Dictyosporium* (*Dictyos. guangdongense* sp. nov. and *Dictyos. variabilisporum* sp. nov.), and one previously documented species (*Dictyos. digitatum*). Specifically, the identification of *Dictyos. guangdongense* is primarily based on its distinct morphology, characterized by complanate, cheiroid, and brown to dark brown conidia, with a hyaline, short, and atrophied appendage arising from the apical cell of the outer row. In addition, the morphological distinctions between *Dictyocheirospora* and *Dictyosporium* are further clarified based on our new data. This study also highlights a few phylogenetic matters regarding *Dictyosporiaceae*.

## 1. Introduction

*Dictyosporiaceae* was established by Boonmee et al. [1] to accommodate a group of cheiroid species, and it is classified within *Pleosporales*, *Dothideomycetes*. The species within *Dictyosporiaceae* exhibit the ability to colonize diverse decaying plant materials in both terrestrial and aquatic habitats, displaying a global distribution [2]. The introduction of novel genera has been significantly increased in the past two decades through extensive sampling and molecular phylogenetic analyses [1,3,4,5,6,7]. Currently, 20 genera are accepted within the family [8], with the majority being recognized as asexual morphs and only 5 genera having been documented to possess sexual morphs, viz., *Dictyosporium*, *Gregarithecium*, *Immotthia*, *Pseudocoleophoma*, and *Verrucoccum* [9,10,11,12,13].

The type genus *Dictyosporium*, typified by *Dictyos. elegans*, was introduced to accommodate the species with sporodochia and cheiroid conidia produced on micronematous conidiophores [1,14]. The generic concept has been amended by Goh et al. [3], and the literature provides a continuously updated compilation of morphological characteristics or keys for all accepted species [3,15,16,17,18]. Subsequently, two sexual morphs from Thailand, viz., *Dictyos. meiosporum* and *Dictyos. sexualis*, and a sexual morph from China, *Dictyos. karsti*, were added to the genus [1,9,19]. The morphological traits of *Dictyosporium* can now be comprehensively characterized, including perithecial, superficial ascomata with an apical ostiole, a membranaceous peridium, cylindric–clavate asci, and fusiform, hyaline, 1-septate ascospores with or without a mucilaginous sheath in its sexual morph, as well as micronematous conidiophores producing cheiroid, olive to brown, complanate conidia composed of 4–7 rows of cells in its asexual morph [1,19]. Currently, 86 species epithets are documented in the genus, and some have been transferred to the similar genus *Dictyocheirospora* based on molecular phylogeny or morphological characteristics [1,20]. Many species have been sequenced, and they cluster in a well-supported monophyletic clade [1,19,21,22,23].

*Dictyocheirospora*, typified by *Dictyoc. rotunda*, is another cheiroid genus that produces pale brown, complanate or non-complanate, and euseptate or distoseptate conidia [1]. The genus contains a total of 28 species, with 8 being transferred from *Dictyosporium*, primarily accomplished by Boonmee et al. [1] and Yang et al. [20]. The new combinations were mainly based on the findings of their phylogenetic analyses. The majority of species have sequence data, and they also cluster in a well-supported monophyletic clade [2,22,24,25]. Four species, viz., *Dictyoc. himachalensis*, *Dictyoc. hydei*, *Dictyoc. indica*, and *Dictyoc. musae*, have not been sequenced yet [20,26].

During our ongoing study of the taxonomy and diversity of freshwater fungi in southern China and along a north–south gradient [27], four interesting collections that morphologically belong to *Dictyosporiaceae* were encountered. Their phylogenetic placements were inferred using a combined dataset of the nuclear ribosomal large subunit 28S rRNA gene (LSU), ribosomal small subunit 18S rRNA gene (SSU), nuclear ribosomal internal transcribed spacers (ITSs), and fragments of the translation elongation factor 1-alpha (*tef1-α*). Three new species are proposed and a new fresh collection is described.

## 2. Materials and Methods

### 2.1. Sampling, Isolation, and Morphological Examination

The specimens were derived from freshwater habitats in Guangdong and Guizhou Provinces, China. The decaying plant materials, including wood, branches, and twigs, submerged in freshwater were collected and placed in zip-lock plastic bags. Essential information such as the collecting site, date, and collector details were recorded. All specimens were transported to the laboratory within three days. If the surfaces of the specimens were heavily covered with mud, they were gently rinsed with tap water. Subsequently, around 4–5 specimens were transferred to a new zip-lock plastic bag or plastic box containing moistened tissue paper/cotton and incubated at room temperature (24–27 °C) for 1–2 weeks. Sampling and specimen incubation followed the method described by Senanayake et al. [28]. A stereomicroscope (Chongqing Optec Instrument Co., Ltd., Chongqing, China) was employed to examine the sporodochia developed on the natural substratum. A compound microscope (Nikon Eclipse Ni-U, Tokyo, Japan) equipped with a digital camera (Canon 750D, Tokyo, Japan) was utilized to capture the fungal structures. Single-spore isolations were made from conidium on potato dextrose agar (PDA, Shanghai Bio-way technology Co., Ltd., Shanghai, China) at room temperature. After aseptic transfer, the germinated spores were incubated at room temperature.

Herbarium specimens (dry wood with fungal colonies) were deposited in the herbaria at Zhongkai University of Agriculture and Engineering (MHZU), Guangzhou, China. Living cultures were deposited in the Zhongkai University of Agriculture and Engineering Culture Collection (ZHKUCC), Guangzhou, China. The novel taxa were registered in the databases Facesoffungi (http://www.facesoffungi.org, accessed on 12 February 2024) [29] and Index Fungorum (http://www.indexfungorum.org/names/names.asp, accessed on 12 February 2024).

### 2.2. DNA Extraction, PCR Amplification, and Sequencing

After a month of cultivation, fungal mycelia were carefully scraped from the colonies for subsequent DNA extraction. The entire genomic DNA was extracted from 100–200 mg of axenic mycelia. The cell fragmentation was accomplished using a homogenizer (Allsheng, Hangzhou, China). The Biospin fungi DNA isolation kit (Bioer, Hangzhou, China) was utilized for genomic DNA extraction in accordance with the manufacturer’s instructions. In this study, primer pairs LR0R/LR5, NS1/NS4, ITS5/ITS4, and EF1-983F/EF1-2218R were used to amplify the LSU, SSU, ITSs, and *tef1-α*, respectively. The amplifications were carried out in a 25 μL reaction volume containing 9.5 μL of double-distilled sterilized water (ddH_2_O), 12.5 μL of 2 × FastTaq PCR Master Mix (Vazyme Co., Nanjing, China), 1 μL of DNA template, and 1 μL of each forward and reverse primer (10 μM). The PCR thermal cycle program for the amplification of the LSU, SSU, ITSs, and *tef1-α* was started with an initial denaturation step at 97 °C for 3 min, followed by 38 cycles consisting of denaturation at 94 °C for 30 s, annealing at 53 °C for 50 s, elongation at 72 °C for 1 min, and a final extension step at 72 °C for 10 min [30,31,32]. PCR products were checked on 1% agarose electrophoresis gels stained with ethidium bromide (EB). The sequencing reactions were carried out by Tianyi Huiyuan Biotechnology Co., Ltd. (Guangzhou, China).

### 2.3. Phylogenetic Analyses

The newly obtained sequences were analyzed using FinchTV v. 1.4.0. The consensus sequences were generated using BioEdit v. 7.2 [33]. The Blast analysis was performed on the NCBI platform (https://blast.ncbi.nlm.nih.gov (accessed on 10 February 2024)) to identify taxonomic matches. The taxa selected for constructing the phylogenetic tree were based on a recent published paper [25] and Blast results obtained from the NCBI. The dataset of each gene region was initially aligned independently using the ‘auto’ strategy (based on data size) by MAFFT v. 7 [34]. Subsequently, manual editing was performed to eliminate uninformative gaps or ambiguous regions in BioEdit v. 7.2 [33]. Each dataset (LSU, SSU, ITS, and *tef1-α*) was concatenated in Mesquite v. 3.81 [35]. The fasta file was converted to PHYLIP (for ML) and NEXUS (for BI) format in the Alignment Transformation Environment (ALTER) online program (http://www.sing-group.org/ALTER/ (accessed on 10 February 2024)). Phylogenetic analyses were conducted with maximum likelihood (ML) and Bayesian inference (BI) algorithms in the CIPRES Science Gateway (http://www.phylo.org/portal2 (accessed on 10 February 2024)) [36]. The ML analysis was performed with RAxML-HPC2 v. 8.2.12 on XSEDE with 1000 rapid bootstrap replicates [37]. The model selected for ML analysis was GTR + GAMMA. The BI analysis was performed in MrBayes v. 3.2.7a [38] and the best-fitting model was estimated via MrModeltest v. 2.2 [39].

The Markov Chain Monte Carlo (MCMC) was run for 10,000,000 generations, and the trees were sampled every 100th generation. The first 25% of the trees that represented the burn-in phase were discarded, and the remaining 75% of the trees were used for calculating the posterior probabilities (PPs) for the majority-rule consensus tree [40]. Phylogenetic trees were visualized using FigTree v. 1.4.0 (http://tree.bio.ed.ac.uk/software/figtree/, accessed on 10 February 2024). The editing and typesetting were accomplished using Microsoft Office PowerPoint 2007 (Microsoft Corporation, Redmond, WA, USA).

## 3. Results

### 3.1. Phylogenetic Analyses

The dataset consisted of the combined LSU, SSU, ITS, and *tef1-α* sequence data of 102 taxa in *Dictyosporiaceae*, with *Periconia igniaria* (CBS 379.86 and CBS 845.96) as the outgroup taxon (Figure 1). The topologies obtained from both the ML and BI analyses exhibited similar patterns across the major clades. The best RAxML tree had a final likelihood value of −24,403.726506. The matrix had 1540 distinct alignment patterns, with 42.54% undetermined characters or gaps. The Bayesian analysis resulted in 24,485 trees after 2,000,000 generations. Phylogenetic analyses indicated that ZHKUCC 24-0001 was classified within the genus *Dictyocheirospora*, exhibiting close relationships with *Dictyoc. gigantica*, *Dictyoc. pandanicola*, *Dictyoc. pseudomusae*, and *Dictyoc. vinaya*. The collection ZHKUCC 24-0004 formed a clade with four strains of *Dictyosporium digitatum* (KUMCC 17-0269, KT 2660, SDBR-CMU459, and yone 280). The collection ZHKUCC 24-0002 clustered with five species of *Dictyosporium*, viz., *Dictyos. alatum*, *Dictyos. appendiculatum*, *Dictyos. pandanicola*, *Dictyos. strelitziae*, and *Dictyos. thailandicum*, while the collection ZHKUCC 24-0003 clustered as a distinct branch within *Dictyosporium* (Figure 1).

### 3.2. Taxonomy

***Dictyocheirospora*** M.J. D’souza, Boonmee & K.D. Hyde, Fungal Diversity 80: 465 (2016).

***Dictyocheirospora submersa*** Y.X. Shu & W. Dong, sp. nov. (Figure 2).

Index Fungorum number: IF901680; Facesofungi number: FoF 15509.

Etymology: referring to submerged wood from which the fungus was isolated.

Holotype: MHZU 24-0001.

Saprobic on decaying wood submerged in freshwater habitats. Sexual morph: undetermined. Asexual morph: hyphomycetes. Sporodochia on natural substratum, punctiform, scattered, and black. Conidiophores reduced to conidiogenous cells. Conidiogenous cells: 3.5–4 × 3–3.5 μm ($\bar{x}$ = 3.8 × 3.3 μm, *n* = 5), holoblastic, integrated, terminal, subcylindrical, and pale brown. Conidia: (20–)35–85 × 12–30 μm ($\bar{x}$ = 57.5 × 22.5 μm, *n* = 30), non-complanate, cheiroid, clavate, ellipsoidal, subcylindrical, consisting of 30–78 cells arranged in 6–7 tightly appressed rows, 4–16 euseptate in each row, slightly constricted at septa, each row inwardly curved, conidial arms appressed, occasionally becoming divergent, with a cuneiform or rounded basal cell, pale olivaceous, brown to dark brown, acrogenous, guttulate, smooth-walled, and without appendages or mucilaginous sheaths. Conidial secession: schizolytic.

Cultural characteristics: Conidia germinating on PDA within 24 h and germ tubes produced from the basal cell. Colonies on PDA reaching 35 mm diam. at room temperature (24–27 °C) in natural light after 20 days, irregular, rough, dry, fluffy, and dense; margins undulate with sparse mycelia, yellow in the middle and white at the margin; and the reverse brown in the middle and pale brown at the margin.

Material examined: China, Guizhou Province, Yuping City, on submerged wood in a river, 7 February 2023, YX Shu, YP2-1.3 (MHZU 24-0001, holotype), ex-type culture ZHKUCC 24-0001. GenBank accession numbers: LSU: PP326216, SSU: PP335106, ITS: PP326193, *tef1-α*: PP333113.

Notes: In the phylogenetic analysis, *Dictyocheirospora submersa* clustered as a distinct branch within *Dictyocheirospora*, exhibiting close affinities with *Dictyoc. gigantica*, *Dictyoc. pandanicola*, *Dictyoc. pseudomusae*, and *Dictyoc. vinaya* (Figure 1). *Dictyocheirospora submersa* can be easily distinguished from *Dictyoc. pseudomusae* by the absence of globose to subglobose, hyaline appendages growing from the apical cells or side of the outer rows [10]. The conidia of *Dictyoc. gigantica* differ from those of *Dictyoc. submersa* by their cylindrical and longer conidia measuring 105–121 × 25–32 μm, whereas *Dictyoc. submersa* has shorter and mostly clavate, ellipsoidal, or occasionally subcylindrical conidia measuring (20–)35–85 × 12–30 μm [3]. *Dictyocheirospora submersa* is quite similar to *Dictyoc. pandanicola* and *Dictyoc. vinaya* in terms of their conidial morphology and dimensions. However, they can be easily distinguished based on the color of their conidia. *Dictyocheirospora submersa* exhibits pale olivaceous, brown to dark brown conidia, while *Dictyoc. vinaya* has reddish-brown conidia (from their photo plate), and *Dictyoc. pandanicola* has pale brown conidia [1,41]. In addition, the conidial arms of *Dictyoc. submersa* are tightly appressed, with minimal divergence observed upon squashing. The conidial arms of *Dictyoc. pandanicola* and *Dictyoc. vinaya*, in contrast, easily become divergent, with the conidial arms of *Dictyoc. pandanicola* even detaching from the conidial body. After conducting a full morphological comparison, we found that *Dictyoc. submersa* does not correspond to any existing species within the genus. Therefore, it is introduced as a novel species of *Dictyocheirospora*.

***Dictyosporium*** Corda, Weitenweber’s Beitr. Nat. 1: 87 (1837).

***Dictyosporium digitatum*** J.L. Chen, C.H. Hwang and Tzean, Mycological Research 95: 1145 (1991) (Figure 3).

Index Fungorum number: IF355284; Facesofungi number: FoF 04487.

Saprobic on decaying wood submerged in freshwater habitats. Sexual morph: undetermined. Asexual morph: hyphomycetes. Sporodochia on natural substratum: punctiform, scattered, black, and granular. Conidiophores: micronematous, reduced to conidiogenous cells. Conidiogenous cells: 3–5 × 3–3.5 μm ($\bar{x}$ = 3.5 × 3 μm, *n* = 5), holoblastic, monoblastic, integrated, determinate, terminal, subcylindrical, and pale brown. Conidia: (20–)30–90 × 20–37 μm ($\bar{x}$ = 57.5 × 28 μm, *n* = 30), complanate, cheiroid, consisting of (30–)50–110 cells arranged in 7–8 tightly appressed rows, two types of shapes: (1) cheirosporous with concave apex, (1–)3–16 euseptate in each row, (2) ovoid or subcylindrical, 10–21 euseptate in each row, slightly constricted and strongly pigmented at septa, yellowish-brown to reddish-brown, dark brown, becoming paler to subhyaline in the terminal top 2–4 cells, terminal cells digitate, straight or incurved, or even curled, hyaline, thin-walled, with a cuneiform basal cell, acrogenous, guttulate, smooth-walled, and without an appendage. Conidial secession: schizolytic.

Cultural characteristics: Conidia germinating on PDA within 24 h, and all the cells can produce germ tubes. Colonies on PDA reaching 30 mm diam. at room temperature (24–27 °C) in natural light after 15 days, irregular, rough, dry, fluffy, and dense in the middle; margin sparse and wavy, white and orange-brown; and the reverse orange-brown in the middle and pale orange-brown at the margin.

Material examined: China, Guangdong Province, Guangzhou City, on submerged wood in a lake, 26 February 2023, YX Shu SDGY4 (MHZU 24-0004), living culture ZHKUCC 24-0004. GenBank accession numbers: LSU: PP326214, SSU: PP335104, ITS: PP326191, *tef1-α*: PP333111.

Notes: In the phylogenetic analysis, the collection ZHKUCC 24-0004 forms a clade with four strains of *Dictyosporium digitatum* (KUMCC 17-0269, KT 2660, SDBR CMU459, and yone 280). The morphological characteristics of ZHKUCC 24-0004, such as possessing cheiroid, reddish-brown conidia with distinctly thin-walled, hyaline, digitate, incurved terminal cells, are in accordance with those of *Dictyos. digitatum* [3,10,41,42]. Therefore, ZHKUCC 24-0004 is identified as *Dictyos. digitatum* based on morphology and phylogenetic analysis. Except for cheirosporous conidia that have been frequently documented in the literature, we have observed the presence of a concave apex in some conidia (Figure 3c–e) from our collection.

***Dictyosporium guangdongense*** Y.X. Shu & W. Dong, sp. nov. (Figure 4).

Index Fungorum number: IF901681; Facesofungi number: FoF 15510.

Etymology: referring to Guangdong Province, from where the holotype was collected.

Holotype: MHZU 24-0002.

Saprobic on decaying wood submerged in freshwater habitats. Sexual morph: undetermined. Asexual morph: hyphomycetes. Sporodochia on natural substratum: punctiform, scattered, black, and granular. Conidiophores reduced to conidiogenous cells. Conidiogenous cells: 4.5–10 × 3–4 μm ($\bar{x}$ = 6 × 3.5 μm, *n* = 5), holoblastic, monoblastic, integrated, determinate, terminal, subcylindrical, and hyaline to pale brown. Conidia: 35–55 × 18–32 μm ($\bar{x}$ = 44 × 23 μm, *n* = 40), complanate, cheiroid, consisting of 50–60 cells arranged in (4–)5(–6) tightly appressed rows, 7–11 euseptate in each outer row, 9–11 euseptate in each inner row, constricted and strongly pigmented at septa, with a cuneiform basal cell, brown to dark brown, acrogenous, guttulate, smooth-walled, with a hyaline, short, wizened appendage arising from the apical cell of the outer row, and without mucilaginous sheaths. Conidial secession: schizolytic.

Cultural characteristics: Conidia germinating on PDA within 36 h and germ tubes produced from the basal cell. Colonies on PDA reaching 50 mm diam. at room temperature (24–27 °C) in natural light after 20 days, irregular, rough, dry, fluffy, and dense; margins undulate with sparse mycelia, white in the middle, with masses of black sporodochia produced at the margin; and the reverse yellowish-brown in the middle and pale yellowish-brown at the margin.

Material examined: China, Guangdong Province, Yangjiang City, on submerged wood in a river, 9 April 2023, YX Shu YJ3-14 (MHZU 24-0002, holotype), ex-type culture ZHKUCC 24-0002. GenBank accession numbers: LSU: PP326213, SSU: PP335103, ITS: PP326190.

Notes: In the phylogenetic analysis, *Dictyosporium guangdongense* forms a clade with *Dictyos. alatum*, *Dictyos. appendiculatum*, *Dictyos. pandanicola*, *Dictyos. strelitziae*, and *Dictyos. thailandicum* (Figure 1). The distinct and well-defined shape of the appendages in *Dictyos. alatum*, *Dictyos. appendiculatum*, *Dictyos. strelitziae*, and *Dictyos. thailandicum* can be identified to distinguish them easily from *Dictyos. guangdongense* [3,9,15,41]. In contrast, the appendages of *Dictyos. guangdongense* are hyaline, short, and atrophied structures arising from the apical cell of the outer row. *Dictyosporium pandanicola* is quite similar to *Dictyos. guangdongense* in terms of their conidial morphology and dimensions. However, *Dictyos. pandanicola* can be distinguished by the absence of any appendages [41]. According to a comprehensive morphological comparison of all species mentioned/not mentioned in the referenced key [3,17,18], *Dictyos. guangdongense* does not match to any known species within the genus. The phylogenetic analyses cannot distinguish between *Dictyos. guangdongense* and its related species mentioned above due to the limited sequence data available for them, which only include LSU and SSU for *Dictyos. appendiculatum*, *Dictyos. Strelitziae*, and *Dictyos. thailandicum*, as well as LSU, SSU, and ITS for *Dictyos. alatum*. The *tef1-α* sequence data are available for *Dictyos. pandanicola*, but not for *Dictyos. guangdongense.*

***Dictyosporium variabilisporum*** Y.X. Shu & W. Dong, sp. nov. (Figure 5).

Index Fungorum number: IF901682; Facesofungi number: FoF 15511.

Etymology: referring to the variable color of conidia of the holotype.

Holotype: MHZU 24-0003.

Saprobic on decaying wood submerged in freshwater habitats. Sexual morph: undetermined. Asexual morph: hyphomycetes. Sporodochia on natural substratum: punctiform, scattered, black, olivaceous, and granular. Conidiophores: 8–22 × 2–4 μm ($\bar{x}$ = 17.5 × 3.5 μm, *n* = 5), semi-macronematous, mononematous, subcylindrical, septate, not constricted at septa, and pale yellowish-brown or pale olivaceous-brown. Conidiogenous cells: 6–9.5 × 2–4 μm ($\bar{x}$ = 7 × 3.5 μm, *n* = 5), holoblastic, monoblastic, integrated, determinate, terminal, subcylindrical, and pale yellowish-brown or pale olivaceous-brown. Conidia: 15–35 × 11–20 μm ($\bar{x}$ = 25 × 16 μm, *n* = 40), complanate, cheiroid, consisting of 12–32 cells arranged in 4–5 tightly appressed rows, (2–)3–7 euseptate in each row, slightly constricted at septa, usually each row has a similar length of cells, with a cuneiform or swollen basal cell, yellowish-brown, producing olivaceous pigmentation at a later stage and becoming olivaceous-brown, acrogenous, guttulate, smooth-walled, and with 2–5 digitate or hypha-like, subcylindrical, hyaline, curved, thin-walled, apical appendages. Conidial secession: schizolytic.

Cultural characteristics: Conidia germinating on PDA within 24 h and germ tubes produced from the basal cell. Colonies on PDA reaching 50 mm diam. at room temperature (24–27 °C) in natural light after 15 days, circular, rough, dry, fluffy, and dense; margin entirely covered with white mycelia; and the reverse pale brown in the middle and white at the margin.

Material examined: China, Guangdong Province, Yangjiang City, on submerged wood in a river, 9 April 2023, YX Shu YJ1-22 (MHZU 24-0003, holotype), ex-type culture ZHKUCC 24-0003. GenBank accession numbers: LSU: PP326215, SSU: PP335105, ITS: PP326192, *tef1-α*: PP333112.

Notes: In the phylogenetic analysis, *Dictyosporium variabilisporum* clustered as a distinct branch within *Dictyosporium* (Figure 1). The complanate, cheiroid conidia with apical appendages of *Dictyos. variabilisporum* correspond to the generic concept of *Dictyosporium*. However, it is unusual within the genus for the variable color of its conidia that are initially yellowish-brown and become olivaceous-brown. In addition, although several phylogenetically related species, viz., *Dictyos. alatum*, *Dictyos. appendiculatum*, *Dictyos. bulbosum*, *Dictyos. krabiense*, *Dictyos. strelitziae*, and *Dictyos. thailandicum*, have apical appendages that are similar to those of *Dictyos. variabilisporum*, they can be distinguished based on the morphology or quantity of their appendages [9,15,41,42,43,44,45]. The earlier stage of *Dictyos. variabilisporum* is also similar to *Dictyos. digitatum* in having digitate, curved, hyaline, thin-walled terminal cells [10,20,41,42]. However, they can be easily distinguished by the conidial morphology, color, and appendages at maturity as shown in Figure 3 and Figure 5. After conducting a full morphological comparison based on the literature and key provided by Goh et al. [3], we found that *Dictyos. variabilisporum* does not correspond to any existing species within the genus. Therefore, it is introduced as a novel species of *Dictyosporium*.

## 4. Discussion

Numerous studies in the early 2,000s on lignicolous fungi in freshwater streams revealed *Dictyosporium* species [46,47,48]; however, the taxa were identified based on morphology, and the names should be questioned. The taxonomy of freshwater fungi and *Dictyosporaceae* has been extensively investigated over the past decades by incorporating barcoding of nuclear ribosomal regions (LSU, SSU, and ITS) and protein-coding genes (*tef1-α* and *rpb2*). The unresolved fungal groups and intriguing taxonomic issues have been effectively addressed using the extensive collections obtained from China and Thailand [49,50,51,52,53]. Furthermore, the continuous generation of a wealth of information has resulted in the publication of numerous reviews, books, and monographs aimed at compiling an updated account of freshwater fungi [24,51,52,53,54,55]. A significant milestone has been achieved by Calabon and his co-authors who provide a comprehensive overview of the different facets of freshwater fungal biology [56].

Within *Dictyosporaceae*, *Dictyocheirospora* and *Dictyosporium* are the two most prominent genera that accommodate the majority of freshwater species, with 11 and 9 reported freshwater species, respectively [24,25]. A geographical distribution review showed that numerous species within the family were reported from China and Thailand [2]. In this study, we describe an additional four hyphomycetes collected from freshwater habitats in China. The findings of our study further validate the role of freshwater habitats as important reservoirs for species of *Dictyosporiaceae*.

*Dictyocheirospora* and *Dictyosporium* are phylogenetically related and share quite similar morphological characteristics. Yang et al. [20] highlighted that *Dictyocheirospora* can be distinguished from the latter genus by its non-complanate or cylindrical conidia, which are predominantly characterized by closely clustered terminal cells at the apex. Based on this classification scheme, *Dictyosporium hydei*, *Dictyos. indicum*, *Dictyos. musae*, and *Dictyos. tetraploides* have been transferred to *Dictyocheirospora* [20]. The confirmation of the transfer through sequence data is still pending; however, we concur with this conclusion based on our newly acquired data. The newly discovered species *Dictyoc. submersa* has non-complanate and subcylindrical conidia with closely clustered terminal cells at the apex (Figure 2), while *Dictyos. digitatum*, *Dictyos. guangdongense,* and *Dictyos. variabilisporum* possess complanate conidia in which the terminal cells are not significantly clustered (Figure 3, Figure 4 and Figure 5).

The phylogenetic relationships between genera in *Dictyosporiaceae* have been well investigated using DNA sequence data [1,2,7,12,24,25]. However, the interspecific phylogenetic relationships in some cases remain ambiguous due to the limitation of insufficient sequence data included in the phylogenetic tree. In this study, the establishment of *Dictyosporium guangdongense* as a novel species is primarily based on its morphological characteristics, as the limited molecular data available cannot distinguish between *Dictyos. guangdongense* and its related species (see notes on *Dictyos. guangdongense*) (Appendix A, Figure 1). In addition, *Dictyosporium aquaticum*, *Dictyos. digitatum*, *Dictyos. palmae*, and *Dictyos. stellatum* exhibit indistinguishable genetic relationships despite their distinct morphologies (Figure 1) [9,15,42,57]. In the phylogenetic analysis, *Dictyocheirospora clematidis* and *Dictyoc*. *thailandica* also cannot be distinguished due to the absence of *tef1-α* sequence data for *Dictyoc*. *thailandica* [25]. The protein-coding genes are quite necessary in species identification within this fungal group.

## Figures and Tables

**Figure 1 jof-10-00259-f001:**
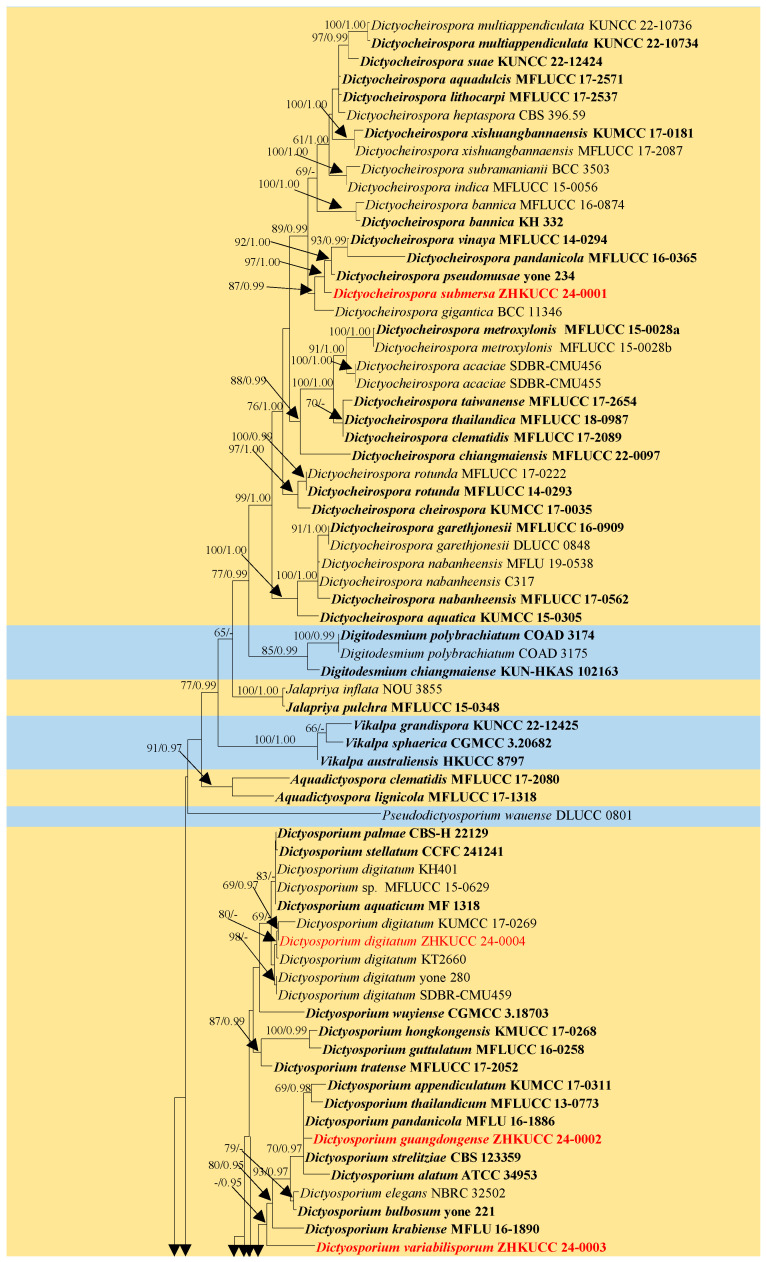
The maximum likelihood (ML) tree is constructed using combined LSU, SSU, ITS, and *tef1-α* sequence data. Bootstrap support values with an ML greater than 60% and Bayesian posterior probabilities (PPs) greater than 0.95 are indicated above the nodes as “ML/PP”. The tree is rooted to *Periconia igniaria* (CBS 379.86 and CBS 845.96). Newly generated sequences are highlighted in red, and the type strains are indicated in bold.

**Figure 2 jof-10-00259-f002:**
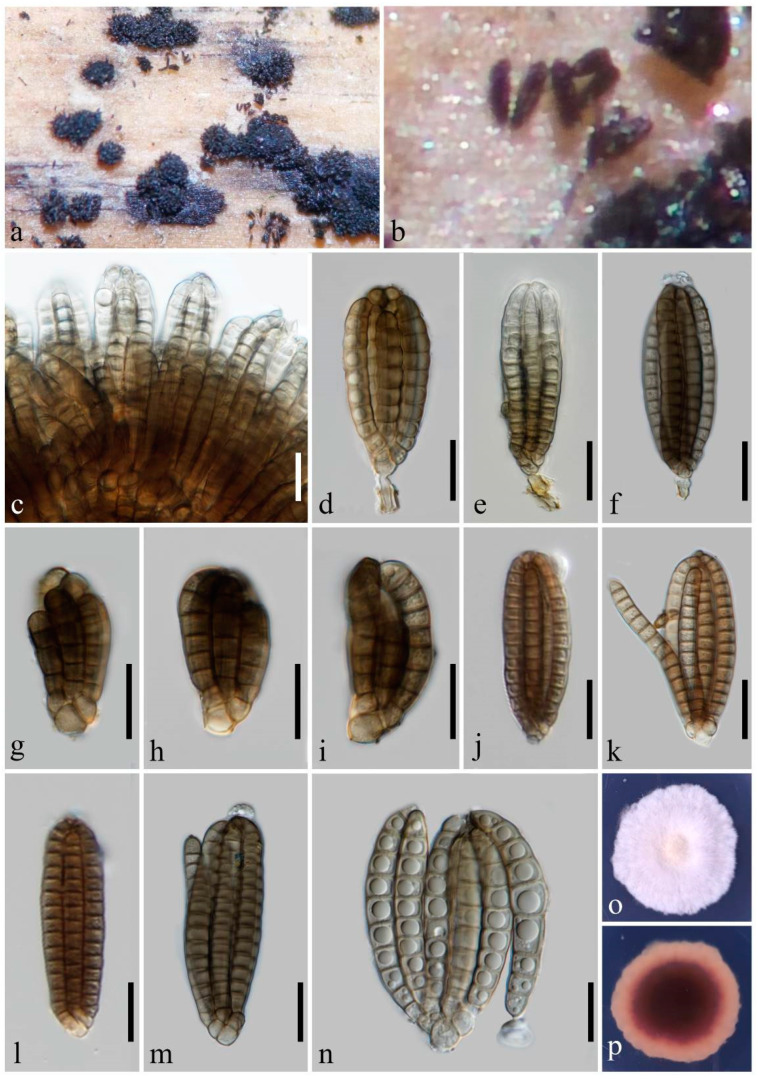
*Dictyocheirospora submersa* (MHZU 24-0001, holotype). (**a**,**b**) Sporodochia on natural substratum. (**c**) Conidia heap. (**d**) Conidiophores with conidia. (**e**–**i**) Young conidia. (**j**–**n**) Conidia. (**o**,**p**) A 20-day-old colony on PDA at room temperature (**o**: obverse, **p**: reverse). Scale bars: (**c**–**n**) = 20 µm.

**Figure 3 jof-10-00259-f003:**
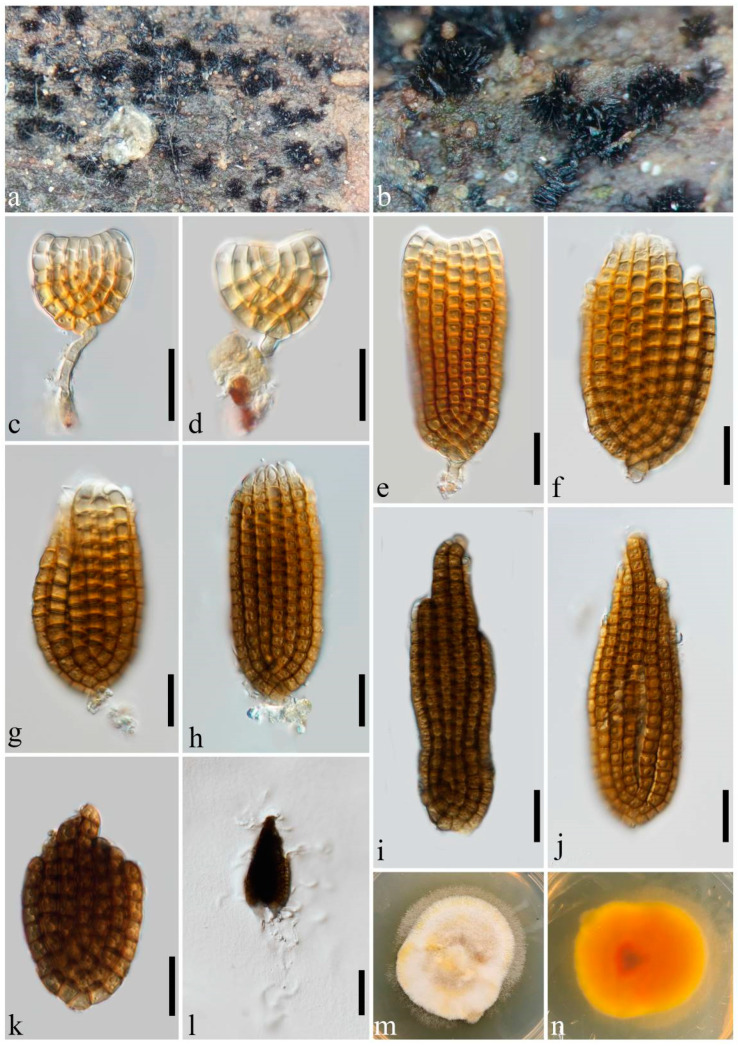
*Dictyosporium digitatum* (MHZU 24-0004). (**a**,**b**) Sporodochia on natural substratum. (**c**–**e**) Conidiogenous cells and young conidia. (**f**–**h**) Conidiophores with mature conidia. (**i**–**k**) Mature conidia. (**l**) Germinating conidium. (**m**,**n**) A 15-day-old colony on PDA at room temperature (**m**: obverse, **n**: reverse). Scale bars: (**c**–**l**) = 20 µm.

**Figure 4 jof-10-00259-f004:**
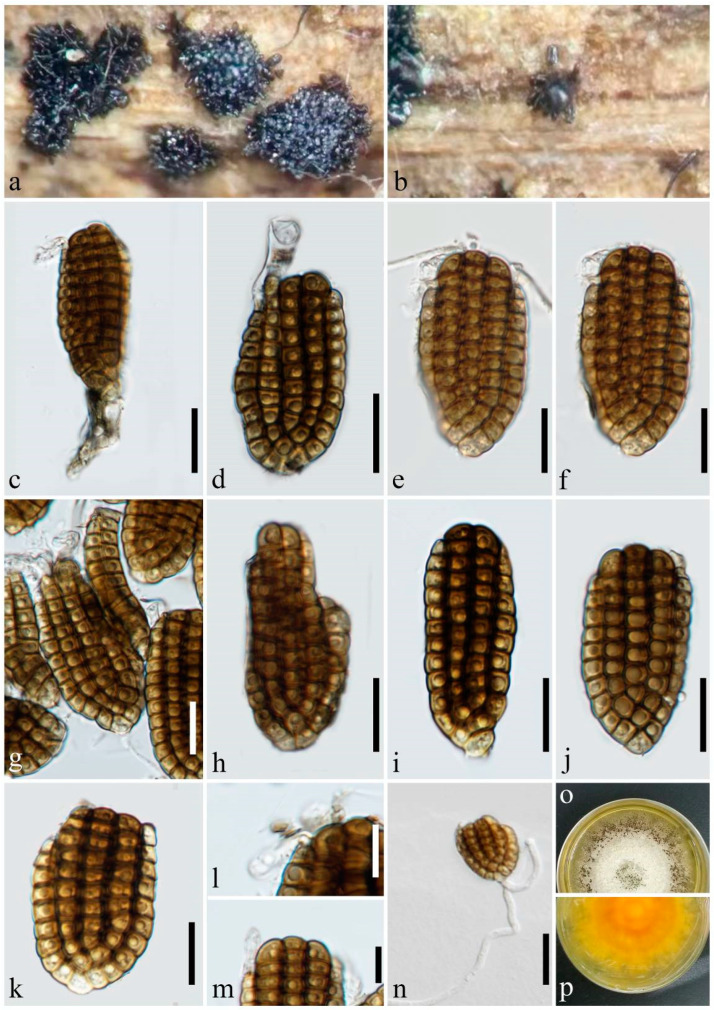
*Dictyosporium guangdongense* (MHZU 24-0002, holotype). (**a**,**b**) Sporodochia on natural substratum. (**c**) Conidiogenous cell with conidium. (**d**–**g**) Conidia with apical appendages. (**h**–**k**) Conidia. (**l**,**m**) Appendages. (**n**) Germinating conidium. (**o**,**p**) A 20-day-old colony on PDA at room temperature (**o**: obverse, **p**: reverse). Scale bars: (**c**–**k**,**n**) = 20 µm; (**l**,**m**) = 10 µm.

**Figure 5 jof-10-00259-f005:**
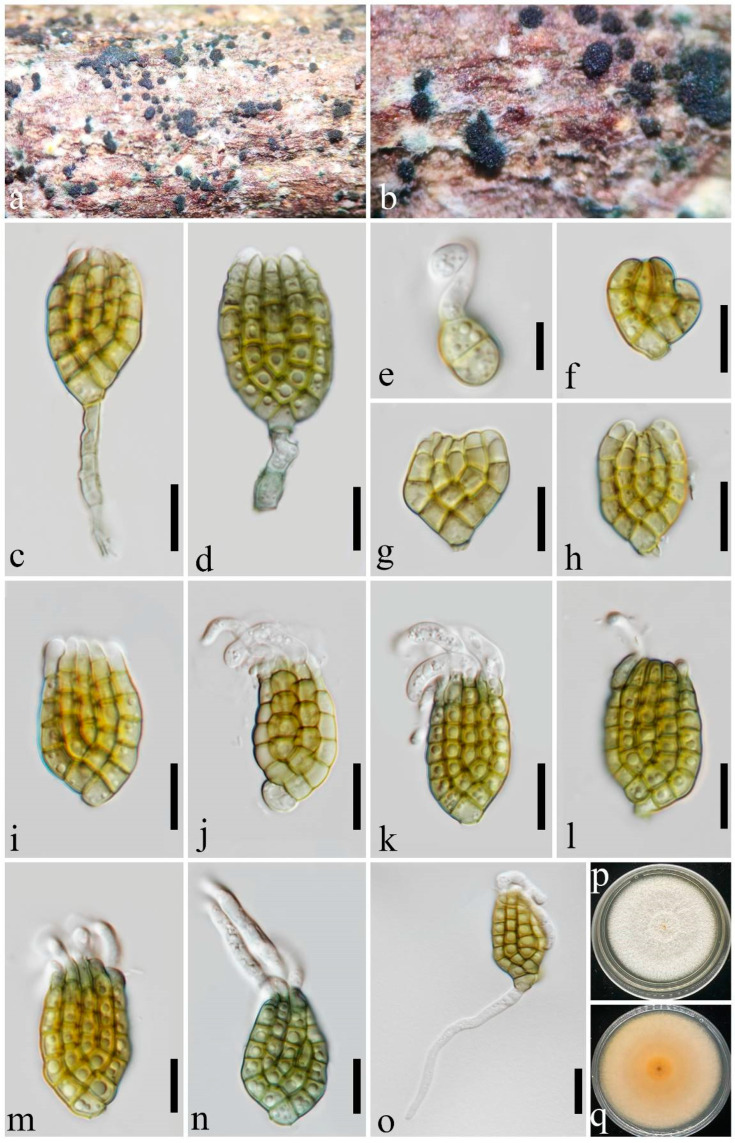
*Dictyosporium variabilisporum* (MHZU 24-0003, holotype). (**a**,**b**) Sporodochia on natural substratum. (**c**,**d**) Conidiophores with conidia. (**e**–**h**) Young conidia. (**i**–**n**) Conidia with appendages. (**o**) Germinating conidium. (**p**,**q**) A 15-day-old colony on PDA at room temperature (**p**: obverse, **q**: reverse). Scale bars: (**c**–**o**) = 10 µm.

## Data Availability

The data generated from this study can be found in the Index Fungorum (http://www.indexfungorum.org/names/names.asp (accessed on 15 February 2024)) and GenBank (https://www.ncbi.nlm.nih.gov/nuccore (accessed on 25 February 2024)).

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
