# Peer review of "Three Novel Cheiroid Hyphomycetes in Dictyocheirospora and Dictyosporium (Dictyosporiaceae) from Freshwater Habitats in Guangdong and Guizhou Provinces, China"

_jof, 2024, doi:10.3390/jof10040259_

Round 1
Reviewer 1 Report
This study enriches our knowledge of the diversity of freshwater ascomycetes of the family Dictyosporiaceae in Chin and fits well with the scope of the Journal of Fungi. The study is focused on Dictyocheirospora and Dictyosporium, in which three new species have been delineated based on sequence analyses of the ribosomal operon (SSU, LSU and ITS) and the protein-encoding gene tef1-α. The species are well supported phylogenetically and have distinct phenotypic features compared with the closely related species of the respective genera. The paper is well organized and written, and the illustrations are of good quality. Only minor comments or corrections are included below.
- All the taxonomic ranks, such as phylum, class, and family, must be in italic. Please review this item in the Abstract and in all the text of the article.
- Avoid the use the same abbreviation for different genera (D.) because it is very confusing. Review all over the text and replace it with a particular abbreviation, e.g., Dictyocheirospora = Dictyoch., and Dictyosporium = Dictyos.
Materials and Methods (M&M)
- L72. Since M&M should not be a text full of cited literature explaining the techniques, please here include a brief explanation on the samplig metodology.
- L74. “The specimens were then incubated in plastic bags lined….” Please give more details. “lined” inside or outside?, tissue paper moistened regularly?
- L75. Replace “ambient temperature” for “room temperature”. If this means that it was the same temperature than that mentioned in the taxonomy section (species description Ls 171, 219, 257, 305), please unify the temperatures, i.e., all “c.a. 25 °C” or “24-27 °C”.
- Ls 98-104. Practically identical to other publications (e.g., Front. Microbiol. 2020 doi: 10.3389/fmicb.2020.00456). Please change a little bit including, for example, PCR procedure.
- Ls108-109. “The taxa selected for constructing the phylogenetic tree were based on recent published papers (Citation)” This sentence requires citation support.
- Ls119. Since the model selected for ML analysis is mentioned in the results (Ls 134 –35), it is not necessary to also mention it here. Consider deleting that from M&M.
Results
- Ls141–143. Consider to expanding here the explanation of the phylogenetic relationships found for the specimens under study. Otherwise, to say they were only identified in two different genera is extremely poor.
- In Figure 1: Put “Dictyosporium digetatum” in italic. Consider replacing the respective numbers “MHZU” (herbarium material) for the specimens under study with those given in the reference fungal collection (i.e., ZHKUCC), which is the same as mentioned in the phylogenetic results (lines 141–143) and other parts of the text.
- Ls172–173 (and for the rest of species descriptions). Delete “; from above” and replace “; from below” by “; reverse”. This should be as follows: “…. sparse mycelia, yellow in the middle and white at the margin; reverse brown in the ….”.
- L220. Replace “from below” by “reverse”.
- In legend for the Figure 2: consider deleting “MHZU 24-0282, holotype” since it is unnecessary because there is only one specimen for this fungus. In “(o, p)” include the incubation temperature and days of the colonies. Replace “from above” by “obverse” and “from below “reverse”. Please do the same in the legends of Figures 3, 4, and 5.
- Ls 232-233. “… we have observed the presence of a concave apex in some conidia (Figure XX) from our collection.” Figure citation for this feature from the plate will be welcome! Delete “new”
- Ls 241. Text with different font size.
- Ls 272-275. The first sentence is incomplete. Consider joining it with the adjacent sentence, i.e. “Based on a comprehensive morphological comparison of all species mentioned/not mentioned in the referenced key [3,17,18], Dictyos. guangdongense does not match to any known species within the genus.” Consider deleting “Therefore, it is introduced as a novel species of Dictyosporium” since it is superfluous.
- L292. Delete “very short”
- L307. The reverse of the colony in not mentioned here. Please consider including it.
- L323. The authors make a mistake mentioning Figure 4, which correspond to Dictyos. guangdongense. However, they are comparing Dictyos. variabilisporum (Figure 5) and Dictyos. digitatum (Figure 3). So, replace the citation of “Figure 4” by “Figure 3”.
Discusion
- L346 “Calabon et al. [57,58]”
- L346 Consider replacing the first part of the sentence after by “Members of the family Dictiyosporaceae exhibit a ….”
- Ls 352-353. Consider replacing the last part of the sentence with “… rare, being only reported in Dictyosporium as mentioned in the introduction.” Since that was already mentioned in the Introduction (L43).
- L354. Delete “the type genus”, not necessary!
- Ls 358-360. Consider deleting all the following sentence “The phylogenetic analyses and morphology indicate that they belong to the genera Dictyocheirospora and Dictyosporium, with three of them representing novel discoveries.”
- Ls 374-384. As a conclusion, some comments on the genus Dictyoscheirospora should also be included in this paragraph.
Reviewer 2 Report
This manuscript describes three new species of Dictyosporiaceae based on collections made in China (Guangdong and Guizhou) and a new report of D. digitatum.
The methods are appropriate. The descriptions are very complete and documented with good pictures of sporodochia and conidia. The phylogenetic tree also corroborate that they are new species, although more data is needed for D. guangdongense as the authors suggest.
This paper could be shortened, some sentences can be more concise and the authors should review the paper to eliminate repetition in the discussion and introduction (these two sections are very similar).
Also, the paper would benefit from more detailed comparisons with morphologically similar species, not only the ones that are clustering with the new species. This may be helpful for people using only morphology to identify Dictyosporiaceae species. For example:
Dictyocheirospora submersa Conidia (20–)35–85 × 12–30 μm (𝑥̅ = 57.5 × 22.5 μm, n = 30), versus
Dictyocheirospora heptaspora Conidia 50–80 × 12–30 μm (𝑥̅ = 57.5 × 22.5 μm, n = 30)
Dictyosporium variabilisporum Conidia 15–35 × 11–20 μm (𝑥̅ _= 25 × 16 μm, n = 40),
Dictyosporium alatum Conidia (22)26–35 × 15–24 μm
Line 16-20:
The abstract needs editing, it should focus on the principal findings; include what is unique for the new species (morphology and phylogeny).
Line 22-25: To facilitate…This part can be left out of the abstract
Line 109: You need a citation. Was it all based on Shen et al 2022, or does it include other databases?
Line 115: Mention if the nucleotide data from the four regions (LSU, SSU, ITS and tef1-α) were concatenated.
Line 137: estimated base frequencies, are they necessary?
Round 2
Reviewer 1 Report
The new version have been improved and the article has merit to be published.
----